# Prebiotics in New-Born and Children’s Health

**DOI:** 10.3390/microorganisms11102453

**Published:** 2023-09-29

**Authors:** Shanmugaprakasham Selvamani, Nidhi Kapoor, Arun Ajmera, Hesham Ali El Enshasy, Daniel Joe Dailin, Dalia Sukmawati, Mona Abomoelak, Muktiningsih Nurjayadi, Bassam Abomoelak

**Affiliations:** 1Institute of Bioproduct Development, Universiti Teknologi Malaysia (UTM), Skudai, Johor Bahru 80000, Malaysia; anazputra@gmail.com (S.S.); henshasy@ibd.utm.my (H.A.E.E.); jddaniel@utm.my (D.J.D.); 2Nutrition Technologies SDN. BHD., No 1 & No 3, Jalan SiLC 2, Kawasan Perindustrian SiLC, Iskandar Puteri, Johor Bahru 80150, Malaysia; 3Faculty of Chemical and Energy Engineering, Universiti Teknologi Malaysia (UTM), Skudai, Johor Bahru 80000, Malaysia; 4Center for Digestive Health and Nutrition, Arnold Palmer Hospital for Children, Orlando, FL 32806, USAarun.ajmera@orlandohealth.com (A.A.); 5City of Scientific Research and Technology Applications, New Burg Al Arab, Alexandria 21500, Egypt; 6Department of Biology, Faculty of Mathematics and Natural Sciences, Universitas Negeri Jakarta, Rawamangun, Jakarta Timur 13530, Indonesia; dalia-sukmawati@unj.ac.id (D.S.); muktiningsih@uni.ac.id (M.N.); 7Orange County Public School, Orlando, FL 32806, USA; mona.abomoelak@ocps.net; 8Specialty Diagnostic Laboratory, Arnold Palmer Hospital for Children, Orlando, FL 32806, USA

**Keywords:** prebiotics, breastmilk, human milk oligosaccharides, formula, weaning, complementary feeding

## Abstract

At present, prebiotics, like probiotics, are receiving more attention as a promising tool for health maintenance. Many studies have recognized the role of prebiotics in preventing and treating various illnesses including metabolic disorders, gastrointestinal disorders, and allergies. Naturally, prebiotics are introduced to the human body in the first few hours of life as the mother breastfeeds the newborn. Prebiotic human milk oligosaccharides (HMOs) are the third largest constituent of human breastmilk. Studies have proven that HMOs modulate an infant’s microbial composition and assist in the development of the immune system. Due to some health conditions of the mother or beyond the recommended age for breastfeeding, infants are fed with formula. Few types of prebiotics have been incorporated into formula to yield similar beneficial impacts similar to breastfeeding. Synthetic HMOs have successfully mimicked the bifidogenic effects of breastmilk. However, studies on the effectiveness and safety of consumption of these synthetic HMOs are highly needed before massive commercial production. With the introduction of solid foods after breastfeeding or formula feeding, children are exposed to a range of prebiotics that contribute to further shaping and maturing their gut microbiomes and gastrointestinal function. Therefore, this review evaluates the functional role of prebiotic interventions in improving microbial compositions, allergies, and functional gastrointestinal disorders in children.

## 1. Introduction

For thousands of years, human evolution has shaped the composition of breastmilk to provide optimal nutrition and protection for the developing neonate at the expense of the mother’s energy supply. Researchers now have a better grasp of the health advantages of breastfeeding due to new insights into human breastmilk composition made via analytical technique development and ‘-omics’ technology integration [1,2]. One of the remarkable features of breastmilk is the diversity and abundance of prebiotics, notably human milk oligosaccharides (HMOs), that are absent in cow milk. HMOs are indigestible to the infant and for this reason reach the colon intact [2,3]. Prebiotics are a class of non-digestible dietary substances that have been shown to promote healthy bacterial growth and/or activity in the gut [4]. It is common knowledge that breastfed infants have a GI microbiota development distinct from formula-fed newborns. At present, prebiotics are receiving much attention, similar to probiotics, as a promising tool in ensuring a healthy state [5,6,7].

Breastmilk is widely acknowledged as the most beneficial source of nutrition for newborns up to the age of 2. However, there are some instances where breastfeeding is still not feasible for a few newborns due to lactation problems, medical reasons, or even personal choice by the mother. Commercial milk formulae are used as an additional supplement to, or as a replacement for (even though not recommended widely), breastmilk in the diets of children and babies aged 0 to 36 months in many countries across the world [8]. Standard infant formula is intended for babies aged 0 to 6 months, whereas follow-up formula is intended for babies aged 6 to 12 months, and toddler formula is intended for babies aged 13 to 36 months [8,9]. In general, infant formula consists of milk proteins, lactose or other carbohydrates, vegetable oils, minerals, and a few additional components. The infant formulas are available to both the very few and the more numerous women who are unable to breastfeed their babies [1,10]. 

Infant formula is often found to lack the beneficial impacts of breastmilk. To date, no commercially available infant formula products are composed of HMOs, which are the third most abundant component of breastmilk after lactose and lipids. Studies have proven that the gut microbiomes of formula-fed infants are less developed and lack bifidogenic effects as compared to exclusive breastfeeding [2,11]. The lack of prebiotics and HMOs in infant formula has been linked, at least in part, to the variations in gut microbiota composition seen between formula-fed and breastfed infants [12,13]. Therefore, numerous studies have been conducted to improve the quality of infant formula [14,15]. Adding commercially available oligosaccharides to infant formula is one way to close the compositional gap between human milk and infant formula [16].

In recent years, the development of biotechnology has allowed for the industrial manufacture of indigestible prebiotics and accelerated research in this area [7]. There are still technological hurdles to overcome, but more and more building blocks of prebiotics are being produced on an industrial scale for the improvement of infant formula. Two major components of HMOs have been successfully proven safe for consumption and have been approved for commercial production [17]. The commercial distribution of these HMOs has led to a better understanding of the role of prebiotics in the early stages of life.

## 2. The Concept of Prebiotics

According to the newly published *Survey on Consumer Insights on Gut Health and Probiotics* by the International Food Information Council (IFIC), around one-third of American adults were found to make efforts to take probiotics regularly [18]. About 60% of those individuals attempt to do so daily and 24% of individuals do so multiple times every day [4]. However, the survey also found that many individuals are unsure of what probiotics and prebiotics are, how they work, and where to obtain them. The poll indicated that around two-thirds of Americans are aware of probiotics, while approximately half are aware of prebiotics. Furthermore, 21% of respondents said they didn’t know how to tell if a meal or drink contains probiotics or prebiotics. Most people consuming probiotic products depend on labels to tell them whether or not such products include probiotics or prebiotics [7,18].

Probiotics are distinct types of living bacteria that, when consumed in sufficient quantities, will be beneficial to the health of the host. In contrast, prebiotics are specific components of foods and supplements that do not undergo human digestive processes [19]. They are metabolized by the human colonic microbiota, which promotes the growth of good bacteria and the development of a healthy gut microbiome. A study carried out by Gibson and Roberfroid [20] first coined the term ‘prebiotic’. They defined prebiotics as the non-digestible food ingredients that selectively stimulate the growth of beneficial microbiota in the human gastrointestinal environment. However, this definition only proposed fructans, such as fructooligosaccharides (FOSs), inulin, lactulose, and galactooligosaccharides (GOSs), as prebiotics [4,20]. In addition, earlier works mainly studied only these elements. The concept of prebiotics was well understood, widely acknowledged, and utilized within the scientific community. However, the original definition has been revised numerous times according to current needs and discoveries without affecting the core values [4]. 

The International Scientific Association of Probiotics and Prebiotics has defined prebiotics as ‘a substrate which is selectively fermented by the gut microflora and bestows health benefits to the host’ [21]. In the early 2000s, a new range of prebiotics were proposed and evaluated for beneficial impacts. These included resistant starches, pectin, and dietary fibers. Currently, the new definition also comprises the non-carbohydrate elements of prebiotics, such as milk oligosaccharides [5,19]. Researchers have looked at the potential health benefits of prebiotics, including the prevention of infections, improvement in cardiometabolic health, increased mineral availability, and modulation of the immune system [6,17,22]. To date, prebiotics such as inulin, FOSs, GOSs, and, more recently, HMOs are the types of prebiotics that have received the most attention [7,12,15]. The fermentable carbohydrates in the gut have the most well-documented health benefits of prebiotic substances. The consensus definition of prebiotics allows for a wide variety of substances that can be targeted to harbor beneficial impacts on different parts of the host’s body [4,7,19].

## 3. Prebiotics in the Early Life of Human 

Several theories have emerged since the discovery that disproved the sterile womb hypothesis, including the possibility that the fetus consumes amniotic fluid, or that phagocytic immune cells move the maternal gut microbiota to her mesenteric lymph nodes and then to her mammary glands [11,23,24]. Many studies supported the theory that newborns receive their microbial loading from their mother’s milk. The newborn’s digestive tract is tolerant and habitable to a variety of bacterial populations, as the immune system is still progressing in development after birth [25]. Both external and endogenous factors allow bacterial colonization to flourish in a controlled and orderly pattern in the infant gastrointestinal environment. From the ages of 2 to 3, more functionally stable microbiota develop, more akin to adult microbiomes [11,25]. 

Although many pathways have been linked to microbial loading, the successful microbial colonization process in the newborn’s gut is highly influenced by human milk. Human breastmilk is regarded as the best diet for newborns. The World Health Organization (WHO) recommends that infants consume only breastmilk during the first six months of life, and then continue nursing for another year while age-appropriate supplementary meals are gradually introduced [26,27,28]. Human breastmilk is regarded as the optimum diet for newborns due to its healthy mix of nutrients and energy. The gut microbial colonization progresses from ‘pioneer’ organisms that are facultatively anaerobic to obligate anaerobes such as *Bifidobacterium*, *Bacteroides*, and *Clostridium* as ‘pioneer’ colonization depletes oxygen in the gut environment [11,29]. 

Several studies have been conducted to understand the impact of breastmilk in shaping the neonate’s gut microbiome [25,27,30,31]. In infants who are fed human milk, the dominance of beneficial microbiota, especially *Bifidobacterium* species, is commonly observed. Additionally, these infants also exhibit colonization by other facultative anaerobic bacteria, albeit to a lesser degree [2,23]. The composition of an infant’s gut microbiota is modulated via the nutritive composition of human milk. As no infant formula can compete with it, human milk is regarded as the undisputed gold standard for nutritional requirements for growing infants and young children [9]. Human milk is a very complex biological fluid that is characterized by the maternal genes and the external factors experienced by lactating mothers [32]. Besides genetic involvement, factors such as maternal diet, lifestyle, health state, pollutant levels, and infant’s age as well and as feeding modes also have a great impact on human milk composition [2]. Overall, human milk fulfills the nutritional, protective, and developmental needs of the growing neonate, tailored to their age-specific requirements, thus ensuring optimal growth and development. [33,34]. 

The well-recognized beneficial impacts of human milk are attributed to a wide range of nutrients and bioactive components. The bioactive components found within breastmilk are anti-oxidative enzymes including lactoperoxidase and superoxide dismutase; growth factors; and immune boosters such as lactoferrin, cytokines, lysozymes, and secretory IgA [2,34]. Metabolites in human milk often change during lactation, from colostrum, which is produced on the first day after delivery, to mature milk at about 4 to 6 weeks postpartum [9,35]. The macro- and micronutrient composition of the milk are also adapted during the transition from colostrum to late lactation for the development of the neonate [36,37]. Recent scientific developments in metabolomic studies have opened windows of opportunity to understand the role and mechanism of how the composition of human milk is reflected by maternal genotype, disease, and lifestyle [26,34,36,38]. 

Human milk also contains non-digestible oligosaccharides or prebiotics–HMOs. The thick and yellowish fluid of colostrum is enriched with about 20–25 g/L of HMOs [37]. Upon maturity, human breastmilk is composed of 5 to 15 g of oligosaccharides in every liter of milk [36,39]. This makes HMOs the third most prevalent solid component of breastmilk, after lactose and milk lipids (Figure 1). 

HMOs are the most prominent components of breastmilk and are distinct bioactive carbohydrates. One of the most useful elements of breastmilk is that HMOs have a complicated structure and several functions [26,40]. These complex sugar molecules have a lactose core at the reducing end. Their differences are based on how they are linked to one or more building blocks such as glucose, galactose, fucose, N-acetylglucosamine or N-acetylglucosamine or N-acetylneuraminic acid residues [2,35,38,41]. There are over a hundred distinct structures of HMOs in human milk (Figure 1). Among them, about fifty types of HMOs are found in significant concentrations. A trisaccharide composed of glucose, galactose, and fucose, 2′-fucosyllactose (2′-FL), is found to be the most common HMO in human breastmilk [37,42]. A recent study quantified the detectable amount of 2′-FL as around 0.06–4.65 g/L of milk samples collected from 400 lactating mothers from 10 different countries [15]. These findings are also supported by work by McGuire et al. [37], where about 65 to 98% of human milk samples obtained in large international cohorts were measured as having 2′-FL at mean values ranging from 0.702 to 3.440 g/L.

Wang et al. [43] demonstrated a comparative analysis of oligosaccharide composition in the milk of several mammals, including humans, goat, sheep, and camels. Based on the findings, the composition of oligosaccharides varied significantly between the different species. For instance, human milk was discovered to have a much greater concentration of fucosylated oligosaccharides [35,37]. The term ‘fucosylation’ refers to the incorporation of fucose monosaccharides into the structure of an oligosaccharide. Fucosylated lactoses, sialylated lactoses, and oligosaccharides with a lacto-N-biose structure such as lacto-N-fucopentaose are some of the most common fucosylated HMOs [19,44,45]. Human milk oligosaccharides are about 80% more fucosylated than bovine milk oligosaccharides, which are around 1–5% [42,46]. In contrast, bovine milk oligosaccharides are mainly composed of sialylated milk. About 35 to 50% of HMOs in mature human breastmilk are fucosylated, whereas about 12 to 14% are sialylated, and 42 to 55% are neutral HMOs that are not fucosylated [43].

## 4. Functionality of Natural Prebiotics in Human Milk

### 4.1. Modulation of Gut Microbial Composition

Several studies have highlighted that the gut microbiota of breastfed newborns is dominated by bifidobacterial strains. Earlier studies often described the ‘bifidogenic’ impact of human milk, referring to the enrichment of beneficial microbiota in an infant’s gut via HMOs [2,23,32,47]. HMOs are resistant to newborns’ digestion as the newborn lacks glycolytic enzymes. Alternatively, HMOs are digested by selected commensal microbial groups residing in the infant’s gut. Members of *Bifidobacterium* genera, including *B. breve*, *B. bifidum*, *B. longum* subsp. *longum*, *B. infantis* and *B. pseudocatenulatum*, have various HMO-degrading enzymes [39,48]. Recent genome investigations on the Bifidobacterial strains isolated from infant guts have revealed that this group of beneficial bacterial genera had a variety of transporters, carbohydrate-binding proteins, and glycosyl hydrolases (GHs) that import and digest HMOs [23,49,50]. James et al. [40] utilized omics studies to elucidate the fucosyllactose metabolic pathway in *Bifidobacterium kashiwanohense*, a novel strain isolated from the feces of breastfed infants. This ability of infant-specific *B. kashiwanohense* to exploit fucosyllactose is a result of selective adaptation of the strain toward the newborn gastrointestinal environment [40].

The discovery of fucose metabolic homologs across the *Bifidobacterium* genus reveals the pathway’s apparent relevance to beneficial microbial colonization in the newborn’s gastrointestinal environment [49,50,51]. Besides *B. kashiwanohense*, other infant-associated *Bifidobacterium*, such as *B. bifidum*, *B. longum* subsp. *infantis*, *B. longum* subsp. *longum*, *B. pseudocatenulatum*, and some species of *B. breve* were also found to have the ability to utilize fucosyllactose as well as other HMO components [39,48]. However, there is still much to learn about how HMOs are specifically metabolized by beneficial microbiota in the newborn’s gastrointestinal environment. Generally, HMOs are transformed by the microbiota into short-chain fatty acids (SCFAs) which contribute to the establishment of stable and healthy gut microbiomes [23,47,52]. The study by Schwab et al. [53] showed that *B. breve*, and *B. longum* subsp. *infantis*, and *B. longum* grown under fucosyllactose supplementation metabolized L-fucose to produce 1,2-propanediol (1,2-PD) via a common L-fucose metabolic pathway. In the gut, 1,2-PD acts as a precursor to propionate via a mechanism that requires glycerol/diol dehydratase as a crucial enzyme [48,53]. This pathway is part of the propionate biosynthesis process. The widespread discovery of genes encoding glycerol/diol dehydratases in the fecal metagenomes of adults revealed that 1,2-PD conversion likely makes a considerable contribution to the synthesis of propionate in the intestinal tract [48]. Production of SCFAs from the HMO’s metabolic pathways suggests the possibility of linkages between the physiology of bifidobacterial and nutritional and health outcomes of infants [23,47].

The production of SCFAs and organic acids via human milk prebiotics can limit the growth of pathogenic microorganisms in the intestinal environment [48,53]. Other members of the gut microbiota, such as *Clostridium*, *Enterococcus*, *Escherichia*, *Eubacterium*, *Lactobacillus*, *Staphylococcus*, *Streptococcus*, and *Veillonella* sp. were found to lack the ability to utilize HMOs entirely or to only use them to a very limited extent [14,54]. A recent study by Salli et al. [55] evaluated microbial alteration using a semi-continuous colon simulator supplemented with 2′-FL. This was indicated by the intermediate synthesis of SCFAs with a lesser production of acetate and lactate found when compared with fermentations in the presence of lactose or GOS [16,55]. The presence of lactose or GOS resulted in higher production of acetate and lactate. The study shows the specificity of 2′-FL as an energy source in this simulated gut model. The study also postulated that the prebiotics in human milk might directly interact with host cells and tissues during the modulation of gut microbiota [55,56].

Therefore, the natural presence of prebiotic components in human milk is mainly for construction, organization, and establishment of a balanced composition of the gut microbiome. HMO metabolism by specific groups of microorganisms and the bifidogenic impacts of the human milk prebiotics contribute to establishing a symbiotic and beneficial community in the growing infant’s gut. 

### 4.2. Enhancing Microbial Adhesion 

Recent research has suggested that prebiotic HMOs may improve the ability of beneficial microbiota to adhere to an infant’s gastrointestinal wall by modulating the expression of bacterial adhesins [57,58,59]. Microbial contact, known as adhesion, occurs between the surface components of bacterial cells entering an infant’s gut, such as lactobacilli, and the surfaces of the host’s digestive tract. The ability of a beneficial bacteria to adhere to the host’s intestinal surfaces has been connected to a wide variety of surface components, including (lipo) teichoic acids, polysaccharides, and proteins [59]. A study by Kong et al. [8] reported that common prebiotic HMOs of 3-FL and LNT2 had enhanced adhesion to the commensal bacterium *Lactobacillus plantarum* strain WCFS1 on Caco-2 intestinal epithelial cells. The study also evaluated the impact of peristaltic shear force on the adhesion process. The presence of prebiotic HMOs and their acid hydrolysis products variably affected glycocalyx-related molecules, antimicrobial peptides, and tight junction gene and protein expression in the presence of *L. plantarum* WCFS1 and shear force. The study reported that shear force exposure increased glycocalyx-related gene expression compared to static incubation in beneficial microbiota [8,60].

Similarly, other recent findings have also reported that prebiotic HMOs boost glycocalyx formation [8,61]. The newborn gut epithelium has a glycocalyx that acts as a binding site for commensal bacteria and a barrier against pathogen adhesion and luminal toxins and enzymes. Proteoglycans are the most vital component that constitute the structural backbone of glycocalyx. Glycocalyx is mostly composed of the glycosaminoglycans heparan sulfate (HS) and hyaluronic acid (HA). There may be an increased risk of gastrointestinal diseases if the glycocalyx components do not mature appropriately in the growing neonates [8,61]. A study reported that supplementation of HMO prebiotics of 3-FL is more effective than 2′-FL in increasing HS and HA, and it also generates greater albumin adsorption. Increases in HS and HA have been found to control mechano-transduction and preserve the integrity of the gut barrier [8,60]. 

### 4.3. Anti-adhesive Strategies against Pathogens

There are multiple ways by which human milk prebiotics are able to protect infants from infectious diseases. As mentioned earlier, prebiotic HMOs are only utilized by selected group microorganisms which makes them a beneficial bacterial group. Pathogenic microorganisms are less capable of utilizing HMOs as energy sources [39,62]. Thus, the growth of beneficial microorganisms was promoted, and this reduced the population of pathogenic microbial groups via competitive exclusion [39,60,63]. In addition, the metabolism of prebiotic HMOs by beneficial microorganisms creates acidic conditions due to the synthesis of organic acids and SCFAs. This limited the growth of pathogenic microorganisms in the infant’s gastrointestinal environment [57,58,59,60]. 

Moreover, HMOs directly prevent the entrance of pathogens by executing the role of soluble receptor decoy. In order to penetrate the host and produce disease, many infectious agents, including bacteria, viruses, and protozoa, must first attach themselves to the glycocalyx. The glycocalyx consists of glycans conjugated to proteins or lipids in the epithelial cell lining [57,59]. Prebiotic HMOs inhibit pathogen attachment to cell surface carbohydrate receptors by acting as carbohydrate-binding ligands or soluble ligand analogs that compete with bacterial adhesins to bind to these receptors, thereby competitively inhibiting their attachment and preventing their colonization. Studies have demonstrated that prebiotic HMOs offer stereospecific protection against a broad range of pathogens. This protection extends to pathogens associated with diarrheal, respiratory, and urinary infections, as well as the human immunodeficiency virus (HIV) [64,65]. Prebiotic HMOs have chemical structures that are similar to glycans, which are used by harmful bacteria to bind to the surface of epithelial cells. Pathogens and toxins that identify and link to HMOs instead of cell-surface glycans will therefore pass through the digestive system without causing infection [65,66].

The anti-adhesive abilities of prebiotic HMOs are dependent on the oligosaccharide charge, and molecular weight, against specific pathogens [60,62,65]. Higher molecular weight prebiotics have been shown to inhibit adhesion of *E. coli* and *Vibrio cholerae*, but not *Salmonella fyris* in the infant gut. In contrast, lower molecular weight prebiotics inhibited the adhesion of *E. coli* and *S. fyris*, but not *V. cholerae*. This indicates the pathogen specificity based on different types of prebiotics. In addition, fucosylation patterns in the prebiotic HMOs also affect the anti-adhesive properties. The prebiotic 2′-FL was found to inhibit the adhesion of several pathogens in the infant’s gut [66,67]. In addition to this, acidic HMOs are also a factor that contributes to the reduction of pathogen adhesion. One research indicated that 3′-SL has suppressed cellular adhesion in uropathogenic *E. coli* (UPEC). *E. coli* serotype O119, *H. pylori*, and *V. cholerae* [63]. 

### 4.4. Antimicrobial and Antiviral Activity

Human breastmilk contains a wide variety of immunological components, which provide a richness of biological activities including protection against diseases caused by bacteria and viruses [63]. Human breastmilk enhances the infant’s first line of defense and has a significant influence on the maintenance of intestinal homeostasis. Most works demonstrating the antimicrobial potential of human milk indicate that, besides maternal antibodies, much of this potential is derived from the microbial composition of the human milk. Various beneficial bacterial strains have been isolated, screened for their antimicrobial properties, and even exploited into commercial products [23,47,52]. However, the antimicrobial properties of human milk prebiotics are still not completely known. Some recent works citing HMOs in antimicrobial activities are focused more on anti-adhesive properties [60,62,66]. 

A recent study has revealed that prebiotic HMOs not only reduced the binding of bacteria but the prebiotics themselves generate an ‘anti-*Campylobacter*’ effect, which reduces the invasion of this pathogenic microorganism [67,68]. The study reported that prebiotic 2′-FL significantly inhibited infection by *C. jejuni* by 80% in human epithelial cell lines of HEp-2 and HT-29 cells. Prebiotics reduced the production of mucosal pro-inflammatory signals such as interleukin (IL) 8 by 60–70%, IL-1 by 80–90%, and the neutrophil chemoattractant macrophage inflammatory protein 2 (MIP-2) by 50% [67]. The study also demonstrated that HMOs prevent the pathogenesis caused by *C. jejuni* in mice [68]. 

In another study, it was demonstrated that heterogeneous HMO extracts possess potent antibacterial and antibiofilm effects against Group B *Streptococcus* (GBS), which is an important neonatal pathogen [62]. Structurally different types of sialylated HMOs were shown to exhibit antibacterial efficacy against GBS in the current investigation. In addition, the study demonstrated that the antibacterial properties of HMO mixes may be produced in vitro by enhancing the cellular permeability of the bacteria. GBS has developed extensive resistance to aminoglycosides, macrolides, and tetracyclines. These results are noteworthy as prebiotic HMOs boost the efficacy of aminoglycosides, lincosamides, macrolides, and tetracyclines, depending on the strain of GBS. Interestingly, HMOs were also discovered to enhance the activity of aminoglycosides against *S. aureus* and *A. baumannii* [62,69]. Another study also demonstrated the antimicrobial effect of prebiotic HMOs against GBS. Using a mouse model, the study discovered that prebiotic HMOs decrease GBS burdens without affecting the vaginal microbiota. The study recommended prebiotic HMOs as a promising alternative to antibiotics for the prevention of GBS perinatal disease [58].

In addition, preterm infants that receive human milk instead of infant formula have a 6- to 10-times lower risk of developing necrotizing enterocolitis (NEC), which is the most frequent and severe intestinal illness. NEC affects 5 to 10% of all very low birth weight newborns [70]. In the United States and Canada, the typical prevalence of NEC in newborns with a birth weight between 500 and 1500 g is around 7%; however, the incidence of the condition in some neonatal critical care units might be significantly higher [14,70,71]. 

Human milk prebiotics are also found to prevent viral infections such as rotaviruses and noroviruses in infants, which cause gastroenteritis epidemics globally. Antiviral properties exhibited by prebiotic HMOs were imitating receptor sites, preventing viral entrance into the cell, and inhibiting viral multiplication within the cell. A study by Koromyslova et al. [72] reported that prebiotic 2-FL can prevent the GI.1 and GII.17 noroviruses from binding to histo-blood group antigens. These data have provided evidence that prebiotic HMOs may serve as a widely reactive antiviral against a number of different norovirus genogroups. A study by Laucirica et al. [73] demonstrated the impact of four types of prebiotics, such as 2-FL, 3′-sialyl lactose (3-SL), 6′-sialyl lactose (6-SL), and GOS, against human rotaviruses type G1P and G2P. Then a study reported that prebiotic 2-FL can reduce infection of G1P, whereas a mixture of 3-SL and 6-SL is the most effective way to inhibit G2P infections [73]. 

### 4.5. HMO Prebiotics in Anti-biofilm Formations

Bacteria tend to adhere to various surfaces, synthesize extracellular polymeric substance (EPS) matrix, form microcolonies, and disperse from the initial surface. This is known as a biofilm. Biofilm production can alter intestinal colonization, infection risk, and antibiotic resistance in infants. Prebiotic HMOs are also evaluated for antibiofilm formation [59,65]. A recent study has demonstrated that the production of biofilms in multi- and pan-drug-resistant *A. baumanii* was significantly decreased in the presence of prebiotic HMOs. These prebiotics primarily exert their effect by inhibiting the production of pellicles, also known as floating biofilms. The same study found that while HMOs were efficient in preventing the formation of new biofilms, they were not able to significantly alter the structure of existing biofilms [57]. This is also supported by another recent study that evaluated inhibition of Gram-positive bacterial biofilm. The study reported that biofilm formation by the *Streptococcus agalactiae* strain was inhibited by prebiotic HMOs to an extent of up to 93%, whereas biofilm formation by methicillin-resistant *Staphylococcus aureus* (MRSA) was inhibited by HMOs to an extent of up to 60% [74]. Biofilm formations by pathogenic bacterial groups affect the efficiency of antibiotics used. It has been demonstrated that the effectiveness of several antibiotics, including clindamycin, erythromycin, gentamicin, and minocycline, against GBS can be enhanced by the presence of HMOs [62].

The antibiofilm activity of entire and fractionated HMO molecules was investigated in recent research on mature biofilms of a variety of clinically relevant Gram-negative and Gram-positive bacteria [59,74]. For Gram-positive bacteria, in particular *E. faecalis* and *S. aureus*, the study found that the number of biofilm cells decreased in a manner that was unique to both the isolate and the strain. On the other hand, HMOs did not demonstrate any substantial effect against Gram-negative bacteria biofilms that had already developed. Although the impact of prebiotics against Gram-negative bacterial biofilm was not observed, the functional significance of these findings may lay in the prevention of skin infections in the breastmilk of mothers and in the nasopharynx of their children [59]. *S. aureus* is well-known to commonly colonize the skin and mucous membranes and has the potential to be readily transferred through nursing. Prebiotic HMOs are capable of inhibiting biofilm formations and thus prevent colonization of pathogenic microbes in the newborn’s gut.

## 5. Prebiotics in Infant Formula

Mothers are encouraged to breastfeed their infants exclusively for the first six months of life. It is important to note that lactation involves much more than providing breastmilk to neonates and young children [9]. The World Health Organization (WHO) recommended that breastfeeding should begin within one hour of birth, and then should be exclusive for the first six months. The lactation should be continued via the introduction of nutritionally adequate and safe complementary foods at six months and should continue up to two years of age or longer. However, many mothers face various challenges that hinder their ability to breastfeed. Infants require milk as their primary sustenance until 12 months of age [1,10,75]. Based on WHO data, many infants and children do not receive optimal nutrients, especially milk, during their early stages of life. The WHO postulates that only about 44% of infants aged between zero to six months worldwide were exclusively breastfed over the period from 2015 to 2020 [28]. Although governments and many other organizers do not encourage it, infant formula is a healthy alternative. All health organizations always prioritize breastfeeding, but formula is always served during the weaning stages, after 6 to 12 months of breastfeeding, or under rare circumstances of insufficient amount of breastmilk production [9,76]. 

The nutritional makeup of human milk is replicated as precisely as possible in infant formula, which is designed specifically for babies. Most baby formulae are either made from cow milk or soy milk, and their goal is to mimic the nutrient profile of human breastmilk [77,78]. Prebiotics in infant formula are modeled after the prebiotic oligosaccharides found in human milk. The most prevalent types of prebiotics included in infant formula are GOSs, FOSs, and/or polydextrose (PDX). There is substantial evidence from clinical studies suggesting that GOSs, which have certain characteristics in common with human milk prebiotics, are advantageous for the health of the digestive tract as well as the immune system [20,79]. Both FOSs and GOSs include lactose at the reducing end; however, FOSs are linear polymers of fructose, whereas GOSs contain lactose at the reducing end and are typically prolonged to six Gal residues, each of which may contain a different branch [7,79].

The FOSs are obtained either via the inverse fructanase and sucrase reaction or the enzyme hydrolysis of inulin. Both pathways result in the production of short-chain FOSs. Long-chain FOSs are produced via the hydrolysis of inulin, which makes up this substance and results in the release of free anomeric carbons and has one fructose [12]. On the other hand, prebiotic GOSs are produced via enzymatic treatment with lactose-β-galactosidase derived from fungi, yeasts, or bacteria [7]. The prebiotic HMOs produced in human milk have a distinct structural makeup compared to prebiotic GOSs and FOSs in infant formula. The addition of GOSs and FOSs as well as inulin is a practical and economical technique to add prebiotic oligosaccharides to infant formula, which ultimately results in an improvement in the formula’s overall quality [13,33,80,81]. It has been agreed upon by the American Academy of Pediatrics (AAP) that the addition of prebiotics to baby formula does not appear to be hazardous to babies who are otherwise healthy. There is still a great deal of study that needs to be done to determine the precise effects of GOS and FOS prebiotics in formula [7,19].

### 5.1. Manufactured Prebiotics (Non-HMO) in Infant Formula

The addition of specified prebiotics to infant formula has a positive influence on the infant’s gut microflora and thus provides advantages similar to breastfeeding. Prebiotic GOSs are the most used prebiotic in infant formula. The inclusion of GOSs in infant formula boosted SCFA production and improved intestinal barrier function in various studies [7,20]. One of the earlier studies, Matsuki et al. [80], demonstrated the bifidogenic effect of GOS-supplemented formula. In a randomized double-blind placebo study, 35 healthy term infants were fed infant formula supplemented with 0.3 g/dl GOSs. Despite the lower supplementation, the inclusion of GOSs in the infant formula significantly increased the population of *Bifidobacterium* and decreased microbial diversity as compared to the control group [80]. A study by Marzorati et al. [82] reported on an in vitro investigation to evaluate the prebiotic effect of GOSs. The administration of the GOS supplement led to an augmentation in the levels of beneficial bacteria, SCFAs, and lactic acid, while concurrently causing a reduction in branched SCFA, pH, and ammonium concentrations [82].

A recent study reported that the inclusion of GOSs into formula inhibited adherence of pathogens to epithelial cells and induced anti-inflammatory and regulatory effects [60]. A study by Bhatia et al. [79] conducted an in vitro investigation showing that GOSs in infant formula had improved intestinal barrier function. This prebiotic was found to upregulate associated gene and protein expression and, therefore, modulate the secretory activity of human goblet cells [79]. Similarly, another study supported the theory that the integrity of the epithelial barrier is reinforced via the capabilities of GOSs, as evidenced by their ability to induce in vitro tight junction assembly in human Caco-2 intestinal cells [83]. In addition, GOSs are also found to have protective effects on gastrointestinal villi structures, which are critical for the undisturbed absorption of nutrients [83]. This is demonstrated via the ability of GOSs in the prevention of deoxynivalenol-induced histomorphology abnormalities in mouse guts. In another study by Akbari et al. [84], GOSs were discovered to have a positive effect on the barrier integrity of Caco-2 cells in vitro. The monolayer integrity was compromised via deoxynivalenol incubation, while GOS incubation protected against this by speeding up the reassembly of the tight junctions. Cellular release of the inflammatory marker CXCL8 was also inhibited by GOS [83,84].

A mixture of GOSs and PDX also showed a bifidogenic effect where the population of probiotics significantly increased in the infant gastrointestinal environment [16]. Prebiotic GOS/PDX supplementation mimicked breastmilk and enhanced the growth of *Bifidobacterium*. The *Bifidobacterium* enhanced acid production, which then favored the enrichment of lactobacilli. A study by Racucci et al. [85] reported the increased abundance of *Bifidobacterium* and *Clostridium* cluster I compared to control studies and also increased protection against respiratory infections. Salminen et al. [86] reported that a group of babies fed infant formula supplemented with GOS-PDX showed higher total lactobacilli, as found in the microfloral profiles of breastmilk-fed babies in the second and third month of infancy. The supplemented GOS-PDX formula had enhanced specific enrichment of *L. delbrueckii* and *L. fermentum* as compared to breastmilk feeding. The study evaluated that the breastmilk origin *L. delbrueckii* was able to utilize supplemented GOS/PDX, which increased their relative abundance as compared to only breastmilk-supplemented groups [86]. 

However, the prebiotics of GOSs and FOSs are the most common and well-established mixture of prebiotics used in infant formula. These prebiotics of GOSs and FOSs are formulated at a ratio of 9:1, respectively, as this proportion mimics the molecular size of HMO distribution in human milk [16,22]. Shahramian et al. [22] reported that babies supplemented with formula with GOSs/FOSs supplementation at a 9:1 ratio for up to 12 months had similar illness history as compared to breastfed children. This indicated that infant formula functions similarly to human milk. The supplemented-formula-fed infants had a shorter diarrhea duration than non-supplemented infants, and these outputs were equivalent to those of breastfed infants. The GOS/FOS-supplemented babies had the same and lower rate of fever and respiratory tract infections, respectively, as breastfed infants, which was lower than those of regular formula-fed infants [22]. 

A study by Grüber et al. [87] evaluated the impact of a similar 9:1 ratio formula with a specific mixture of short-chain GOSs, long-chain FOSs (lcFOSs) (6.8 g/L, ratio 9:1), and pectin-derived acidic oligosaccharides (1.2 g/L). This formulation was found to reduce atopic dermatitis by 44% in infants not at risk in their first year. After stopping oligosaccharide administration, this substantial impact was lost at preschool age [87]. A recent ELFE cohort study evaluated the impact of prebiotic as well as probiotic inclusion in the formula among 8389 children from age 2 to 10 months old [6]. Their study found that there was no significant correlation between supplementation with GOS or FOS, either individually or combined, in preventing respiratory illness. However, consumption of these prebiotics from the early stages of life has been proven to significantly reduce upper respiratory tract infections as compared to non-supplemented control groups [6]. 

A recent study by Neumer et al. [88] assessed the effects of an infant formula enriched with a mixture of prebiotic short- and long-chain inulin-type oligosaccharides (FOSs) on health outcomes, safety, and tolerance. The addition of prebiotic Orafti^®^Synergy1 to a regular infant formula was accepted well by the babies and marginally improved their well-being, as measured by numerically reduced daily crying times compared to control groups. This study also validated earlier findings that the inclusion of prebiotic FOSs in baby formula had boosted *Bifidobacterium* populations in the fecal microbiome [88]. In addition, Zhu et al. [89] demonstrated the impacts of prebiotics and organic polyphenol (OPO)-supplemented infant formula to produce feeding results comparable to those associated with human milk. The study reported that there was no significant difference in the alpha diversity of the gut microbiota between the groups that were OPO and those that were nursing with breastfeeding. The relative abundance of *Enhydrobacter* and *Akkermansia* in the standard infant formula was comparable to those of infant formula supplemented with the OPO group. The analysis of gut microbiota metabolic functions via urinalysis also indicated that formula supplemented with OPO produce results similar to those obtained in breastfeeding groups [89]. 

The addition of prebiotic components to formula has proven to generate beneficial impacts in infants. Prebiotics are naturally present in breastmilk, and infants on supplemented formula have a lower stool pH, better stool consistency and frequency, and a higher concentration of bifidobacterial in their intestine compared to infants on a non-supplemented standard formula These findings emphasize that formula must be similar to human milk composition and must elicit effects comparable to human milk. 

### 5.2. HMO Prebiotics in Infant Formula

Prebiotic HMOs are characterized by their extremely complicated structural makeup as well as their broad range [35,39]. As a result of this, there is a lack of available data regarding the use of comparable structures in infant formulae. This was the reason for the usage of a prebiotic combination that consisted of 90% short-chain GOSs and 10% long-chain FOSs in infant formula [9]. The purpose of this mixture is to emulate the impact that human milk has on the gut microbiome as a prebiotic [7]. For almost two decades, infants’ nutritional products have been supplemented with complex prebiotic oligosaccharides that are derived from non-human sources rather than those derived from humans [7,90]. Even though there are many pre-clinical and clinical studies that have been conducted to evaluate the characteristics of GOSs and FOSs, their positive benefits are yet unknown [15,27]. After further investigation, the EFSA Panel on Dietetic Products, Nutrition, and Allergies concluded that there is no compelling need to include GOSs and FOSs in baby formula [91]. It is not difficult to develop an infant formula with HMOs. However, the production of HMOs in large quantities is a laborious and expensive procedure that yields inadequate amounts for clinical applications, as the process is not automated.

Historically, direct extraction of prebiotic HMOs from actual human milk was well established. Several methods have been described, such as solid phase extraction; centrifugal separation; ultrafiltration; liquid extraction; and gel filtration chromatography [10]. However, these methods are not appropriate for the manufacturing of HMOs in a commercial setting. Then, Glycom A/S became the world’s leading HMO supplier and developed HMO synthesis through chemical processes between the years 2005 and 2012 [92,93]. The manufacturer had developed substantial quantities of 2′-FL and lacto-N-neotetraose (LNnT) for use in pre-clinical and clinical research programs. In September 2015, the United States approved the usage of two chemically synthesized HMOs, 2′-FL and LNnT [92]. A significant number of complex HMO structures such as 2′-FL, 3-FL, DFL, LNT, LNnT, LNFP I, and LNFP III were commercially produced via chemical synthesis pathways [69]. Recently, Bandara et al. [94] have devised a convergent synthetic technique for the chemical synthesis of LNnH. However, the chemical synthesis of HMOs is prohibitively expensive, and their process complexity has restricted their availability [92,95].

On the other hand, Abbott Laboratories and Kyowa Hakko Kogyo Co. have utilized the biosynthetic capability of microbial cells by coupling two or more separate cell homogenates. As a result, they were able to create higher quantities of prebiotic LNnT and fucosylated oligosaccharides. This was accomplished by coupling two or more distinct cell homogenates [69,96]. Additionally, chemo-enzymatic synthesis has also been utilized for the production of HMOs. In this process, glycosyltransferases are first recombinantly produced, and then they are used in conjunction with nucleotide-activated donor substrates and acceptors that match [92,93,97,98]. To date, researchers have widely utilized N-acetyl-glucosaminyltransferases, fructosyltransferases, sialyltransferases, and glycosyltransferases for the enzymatic production of HMOs [93,99]. 

A study by Zeuner et al. [99] showcased the successful synthesis of LNnT via the utilization of glycosidases derived from natural sources. In this study, the researchers utilized the enzymatic activity of β-galactosidase derived from *Bacillus circulans* to introduce lactose moieties onto lacto-N-triose acceptor molecules [99]. Similarly, another study utilized LNB phosphorylase from *Bifidobacterium longum* subsp. *infantis* to construct β-1,3-linked neutral HMOs, including LNB and LNT [97]. However, the chemo-enzymatic synthesis platform is not capable of producing complex HMOs in sufficient numbers for commercial use. This synthetic platform is only able to produce HMOs in milligram quantities [92,98].

The availability of commercially generated HMOs is growing, and because HMOs have therapeutic features, it is possible that they may be used to improve health and treat illnesses that are not directly related to their recognized activities in newborns. Among several structurally significant HMOs available for synthetization, the prebiotic 2′-FL and LNnT are present in goods for newborns in over 30 countries, and there is a growing presence of these substances in non-infant products [45,93]. There have been a restricted number of intervention studies conducted on newborns in order to evaluate the safety and effectiveness of baby formulas enriched with HMOs. Table 1 summarizes the research on infant formula fortified with 2′-FL and LNnT. 

Based on Table 1, all reported studies have proven that supplementation with prebiotic 2′-FL is safe for infants [12,45,100,101]. A clinical trial by Goehring et al. [101] was the first work to demonstrate the impact of 2′-FL on the development of the immune system in newborns. The study reported that infants fed with 2′-FL supplementation displayed innate cytokine profiles that were positioned between breastfed and regular formula feed (GOS only), resembling the profiles of breastfed infants to a greater extent. Even a lower dosage of 2′-FL (0.2 g/L) is able to modulate cytokine profiles in relation to breastfed babies [101].

**Table 1 microorganisms-11-02453-t001:** Clinical interventions using synthetic HMOs.

Reference	Interventions	Participants	Duration	Outcomes
Marriage et al. [100]	Control: breastfed Treatment: 3 types of formula [(a) GOS 2.4 g/L; (b) GOS 2.2 g/L + 2′-FL 0.2 g/L; and (c) GOS 1.4 g/L + 2′-FL 1 g/L]	Control (n = 65) and treatment (n = 189)	4 months	Formulas supplemented with 2′FL are well tolerated, and 2′FL absorption profiles are similar to those of breastfed infants.
Goehring et al. [101]	Control: breastfed Treatment: 3 types of formula [(a) GOS 2.4 g/L; (b) GOS 2.2 g/L + 2′-FL 0.2 g/L; and (c) GOS 1.4 g/L + 2′-FL 1 g/L]	Formula-fed (n = 317) or breastfed (n = 107)	4 months	2′-FL exhibits lower plasma and ex vivo inflammatory cytokine profiles, similar to those of a breastfed reference group
Kajzer et al. [12]	Control: breastfed Treatment: 2-types of formula [(a) No oligosaccharides; (b) short chain FOS 2.0 g/L + 2′-FL 0.2 g/L]	Breastfed (n = 43); no prebiotics (n = 42); and with prebiotics (n = 46)	4 months	2′FL with FOSs demonstrated good tolerance, consistent stool consistency, formula intake, anthropometric measurements, and percentage of feedings resulting in spit-up or vomit. These findings were comparable to infants who were fed formula without oligosaccharides or human milk
Steenhout et al. [102]	Control: breastfed and cow milk-based infant formula (no oligosaccharides) Treatment: cow milk-based infant formula + 2′-FL 1.0 g/L + LNnT 0.5 g/L	Breastfed (n = 38); no prebiotics (n = 87); and with prebiotics (n = 88)	4 months	2′-FL and LNnT shift the stool microbiota and metabolic signature toward those observed in breastfed infants
Puccio et al. [13]	Control: infant formula (no prebiotics) Treatment: infant formula + 2′-FL 1.0 g/L + LNnT 0.5 g/L	Control (n = 87); treatment (n = 88)	6 to 12 months	Infant formula supplemented with 2′FL and LNnT is safe and well tolerated. Decrease in respiratory tract-related morbidity outcomes in infants fed with formula containing 2′-FL and LNnT
Storm et al. [103]	Control: formula + *B. lactis* only Treatment: formula + B. lactis + 2′-FL 0.25 g/L	Control (n = 33); treatment (n = 30)	6 weeks	Partially hydrolyzed infant formula with 2′FL and B lactis is tolerated well
Berger et al. [33]	Control: Breastfed and infant formula (No oligosaccharides) Treatment: infant formula + 2′-FL 1.0 to 1.2 g/L + LNnT 0.5 to 0.6 g/L replacing equivalent lactose	Breastfed (n = 35); no prebiotics (n = 63); and with prebiotics (n = 58)	6 to 12 months	Shifts the stool microbiota and metabolic signature toward those observed in breastfed infants; increased in bifidobacterial population, less require antibiotics
Dogra et al. [49]	Control: infant formula (No prebiotics but with 1.5 g/L additional lactose) Treatment: Infant formula + 2′-FL 1.0 g/L + LNnT 0.5 g/L	Control (n = 103); treatment (n = 106)	6 to 12 months	Reduced risk for reported bronchitis and lower respiratory tract illnesses
Parschat et al. [104]	Control: breastfed and infant formula (no prebiotics) Treatment: Infant formula + mixture of 5-HMOs (2.99 g/L 2′-FL, 0.75 g/L 3-FL, 1.5 g/L LNT, 0.23 g/L 3′-SL, and 0.28 g/L 6′-SL)	Breastfed (n = 116); No prebiotics (n = 112), and with prebiotics (n = 113)	4 months	5HMO-Mix at 5.75 g/L in infant formula is safe and well tolerated by healthy term infants during the first months of life
Alliet et al. [17]	Control: breastfed Treatment: (a) formula + *L. reuteri*; and (b) formula + 2′-FL 1 g/L]	Breastfed (n = 60); treatment (n = 289)	6 months	2′-FL supplementation shifted gut microbiota composition similarly to breastfeeding, but no significant weight gain as compared to probiotic-supplemented milk. The study suggests that 2′FL has incremental effects on top of *L. reuteri* in infant formula
Bosheva et al. [105]	Control: breastfed and cow milk-based infant formula (No prebiotics) Treatment: (a) formula + 1.5 g/L 5-blended HMOs and (b) formula + 2.6 g/L 5-blended HMOs (2′FL, DFL, LNT, 3′SL,6′SL)	Breastfed (n = 35); no prebiotics (n = 63), and with prebiotics (n = 58)	6 months	Increase in the relative abundance of the Bifidobacterial group as compared to controls; SCFA production close similar to breastfed; prebiotics supported the development of the intestinal immune system and gut barrier function
Gold et al. [77]	Control: amino acid-based formula (AAF) (no HMOs) Treatment: AAF + 2′-FL 1.0 g/L + LNnT 0.5 g/L	Cow’s milk protein allergy (CMPA) infants with no breastfeeding; control (n = 32); treatment (n = 29)	4 to 12 months	Supplementation with 2′-FL and LNnT was associated with significant enrichment in HMO-utilizing bifidobacteria and a partial correction of the gut microbial dysbiosis in infants with CMPA
Vandenplas et al. [106]	Control: whey hydrolyzed formula (no HMOs) Treatment: whey hydrolyzed formula + 2′-FL 1.0 g/L + LNnT 0.5 g/L	Control (n = 96); treatment (n = 94)	4 months	Reduction in relative risk of lower respiratory tract and gastrointestinal infections
Lasekan et al. [107]	Control: breastfed and infant formula (No prebiotics) Treatment: infant formula + mixture of 5-HMOs (5.75 g/L; 2′-FL, 3-FL, LNT, 3′-SL and 6′-SL)	Breastfed (n = 104); No prebiotics (n = 129), and with prebiotics (n = 130)	4 months	Formula containing five HMOs supported normal growth, gastrointestinal (GI) tolerance, and safe use in healthy term infants
Boulangé et al. [78]	Control: whey hydrolyzed formula (no HMOs) Treatment: Whey hydrolyzed formula + 2′-FL 1.0 g/L + LNnT 0.5 g/L	Cow’s milk protein allergy (CMPA) infants with no breastfeeding; Control (n = 97) Treatment (n = 97)	4 months	Enriched microbiome with HMO-utilizing bifidobacteria and slowed the progression of the microbiome composition toward an adult-type pattern. HMO supplementation partially reversed the dysbiosis in infants with CMPA and shifted the microbiome composition closer to a pattern typical of breastfed infants
Holst et al. [81]	Control: treatment: formula + 5HMO-mix 5.75 g/L (2′-FL,3-FL, LNnT,3′-SL,6′-SL)	Treatment (n = 113)	4 months	Shifts the infant fecal microbiome closer to that of breastfed infants; decreases the population of opportunistic pathogenic strains down to the level observed in breastfed infants during the first 4 weeks
Hill et al. [108]	Control: breastfed Treatment: 3 types of formula [(a). Control: no HMOs: GOS 2.4 g/L; (b) low dosage: GOS 2.2 g/L + 2′-FL 0.2 g/L; and (c) high dosage: GOS 1.4 g/L + 2′-FL 1 g/L]	Breast-fed (n = 51); control—no HMOs (n = 48); low dosage (n = 54); and high dosage (n = 48)	4 months	2′-FL inclusion resulted in significant increases in serum metabolites derived from microbial activity in the gastrointestinal tract and; an increase in bile acid production; 2′-FL supports the production of secondary microbial metabolites at levels comparable to breastfed

A study by Steenhout et al. [102] and Puccio et al. [13] was among the earlier studies that evaluated infant formula enriched with a combination of 2′-FL at a and LNnT (Table 1). Both studies demonstrated that prebiotics are safe and well tolerated for infant growth. The findings by Puccio et al. [13] indicated a decrease in respiratory tract-related morbidity outcomes in infants fed with infant formula containing 2′-FL and LNnT compared to those who were administered the control formula [13]. A recent study by Alliet et al. [17] compared infant formula supplemented with probiotics only and another with prebiotics only, with reference to breastfed infants. This recommended that infant formula supplemented with 2′-FL has incremental effects on top of infant formula supplemented with *L. reuteri* and may help move infant formula-fed newborn gut microbial patterns closer to breastfed infants [17].

A study by Bosheva et al. [105] reported on clinical trials using 5 types of HMOs blended with infant formula at two different concentrations. The concentrations of the five HMOs in treatment studies were in the range of that reported in human milk for the individual HMOs. In total, these mixtures of HMOs were added to the infant formula at 1.5 g/L and 2.5 g/L. This special combination of five HMOs in infant formula resulted in improved intestinal immune development and gut barrier function. The prebiotic-enriched infant formula altered the gut microbiota closer to that of breastfed infants with increased bifidobacteria, notably *B. infantis*, and reduced toxigenic *Clostridioides difficile* [105].

As the first regulatory approval was given to chemical synthesis in 2015, Glycom A/S obtained approval for their microbial fermentation in November 2016. Jennewein received regulatory approval in 2015, whereas Glycosyn/Friesland Campina Domo and Inbiose/DuPont were authorized as HMO producers in 2018 [14,95]. For recent developments, GeneChem obtained regulatory approval from the FDA for their chemo-enzymatic synthesis of 3′-sialyllactose (3′-SL). Glycom’s chemically synthesized 2′-FL and LNnT were the first two HMOs approved in the EU, followed by microbially fermented versions [45]. These authorizations led to the worldwide commercialization of prebiotic HMO 2′-FL and LNnT, mostly in infant formula but also in dietary supplements and medicinal foods [14,95].

In June 2015, the European Food Safety Authority (EFSA) conducted a favorable evaluation of 2′-FL [44]. This assessment was conducted using scientific and technical data, and the EFSA announced the findings accordingly. EFSA declared 2′-FL safe for babies up to one year old. This was carried out in conjunction with LNnT at doses not exceeding 1.2 g/L of 2′-FL and 0.6 g/L of LNnT. The ratio of 2′-FL to LNnT in the reconstituted formulas was maintained at 2:1. According to research, it has been determined that the inclusion of 2′-FL in follow-on and young-child formulas, at concentrations not exceeding 1.2 g/L of 2′-FL (either alone or in combination with LNnT, at concentrations not exceeding 0.6 g/L, at a ratio of 2:1), is considered to be safe for young children who are older than 1 year of age [41,44,45].

In conclusion, newborns fed EFSA-approved prebiotic HMOs have shown positive responses similar to breastfed newborns. Further research is always needed to validate the long-term impacts of prebiotic HMO on newborns, focusing more on gut microbiome, immunity, and other relevant criteria. About 20 to 30% of European mothers had been found to lack oligosaccharides such as 2′-FL in their milk. Studies on non-secretory mothers’ breastfed children’s infection risk have shown pros and cons. Fucosylated oligosaccharides may improve or harm baby formula. However, the presence of individual oligosaccharides in human milk does not justify the anticipated advantage of structurally similar synthetic ones in infant formula. Human milk oligosaccharides possess intricate and distinctive characteristics, and their impact on infant health remains uncertain. The complexity of human milk oligosaccharides poses challenges for incorporation into baby formula and the utilization of synthetic oligosaccharides in infant formula lacks sufficient data to support their widespread adoption. 

## 6. Prebiotics during Weaning and Complementary Feeding

Weaning is the process of the gradual reduction of breastfeeding to newborns alongside the subsequent introduction and increasing feeding of other foods and fluids [90,109]. The WHO defines this transition from milk-based feeding to solid-type foods for newborns as ‘complementary feeding’ [28]. Babies’ growth and development are profoundly influenced by their diets, with optimal growth and health depending on the mother’s ability to provide a nutritious diet. Various international organizations, including the World Health Organization, the European Food Safety Authority (EFSA), and the American Academy of Pediatrics (AAP) have always paid close attention to infant formula as well as complementary feeding [110,111,112]. For instance, minimum and/or maximum limitations are set for certain nutritional components of food including proteins, carbohydrates, fats, minerals, and vitamins. It is crucial for parents to learn to identify baby meals as distinct from other foods that may be promoted as such but are not actually appropriate for newborns owing to their composition or safety [28,90].

The time to start weaning is based on individual preference, which can be impacted by the health state of the mother for continued lactation, personal lifestyle, postpartum effects, and so on [28]. According to the guidelines set out by the American Academy of Pediatrics (AAP), it is recommended that newborns be exclusively fed with breastmilk for a duration of 6 months following their birth. Then, it is recommended that infants be introduced to a combination of solid meals and breastmilk for a duration of at least 6 months, continuing until they reach the age of 1 year [111]. However, this has been disputed by more recent research, which suggests that the timing of solid food introduction should instead depend on factors such as an infant’s oral development, nutritional needs, and exposure to environmental disease load [113]. Additionally, another cohort study revealed that short duration of breastfeeding, which is 4 months or less, was often associated with an increased risk for being overweight during childhood, rather than the early introduction of solid foods, and the risk is not different between breastfed and formula-fed infants [114].

Breastmilk has important advantages beyond its nutritional content, including substances that modulate the immune system in newborns. Following the process of childbirth, breastfeeding plays a crucial role in facilitating the establishment and development of the newborn gut microbiota. The composition of bacteria present in breastmilk has also been reported to differ across several stages of lactation, ranging from colostrum to late lactation. Additionally, the chemical composition of the breastmilk also altered during similar time frame. The transition in the microbial diversity and composition of the breastmilk naturally occurred to modulate and establish a mature gut microbiota in growing infants [41]. Infant formula was invented to fulfill the role of breastmilk in establishing a mature gut microbiome, promote brain development, and boost the development of the immune system in growing infants. [95,115]. Thus, there are different staged and follow-on formula products, which exhibit variations in composition based on the necessary daily allowances and the introduction of complementary meals [116]. Undoubtedly, exclusive breastfeeding remains the gold standard of nourishment for infants, and numerous studies indicate that this should be extended at least to 1 year or longer, as mutually desired by the mother and infant.

An infant’s capacity to digest carbohydrates is constrained primarily to simple carbohydrates such as lactose and sucrose rather than complex carbohydrates [41]. During the weaning process, quantities of salivary α-amylase and pancreatic α-amylase are shown to be lower compared to those observed in adults. The WHO still recommends complementary feeding, which must be accompanied by breastfeeding. Early weaning can lead to a reduction in the diversity of the gut microbiome, which can affect the development of the immune system and increase the risk of digestive problems. Continuous breastfeeding has been proven to smoothen the transition of infants’ diets from milk-based to containing solid foods. This clarifies the importance of continuing supplementation of prebiotics in infants even after the introduction of solid foods [28,110,111,112]. Therefore, during the weaning process for non-breastfeeding infants, supplementation of prebiotics in solid foods must be continued to establish and maintain a healthy gut microbiome in children.

Extensive studies have been carried out on how breastmilk and formula affect the microbiome and health of babies. Recent developments have initiated efforts to mimic HMO composition in breastmilk into infant formula [12,17,100,101]. However, there are fewer clinical studies evaluating the role of prebiotics during weaning or complementary feeding. Most of the recommendations are more on the introduction of ‘solid foods’ to infants as similar to adults. The process continues with gradual introduction until it reaches a diet completely similar to an adult’s in one to two years. Prebiotics such as HMOs and other non-digestible oligosaccharides including GOSs and FOSs have a significant impact on the community structure and metabolic processes of the microbiota that lives in an infant’s digestive tract [16]. The effort to impart similar bifidogenic effects on formula-fed infants must be continued during the weaning process.

A study by Hugenholtz et al. [117] evaluated the transition that occurred during weaning, where the single breastfed baby was followed from birth to six months of age, during which time formula, dairy, and solid foods were introduced. This meta-transcriptomic analysis of fecal samples revealed that the beta-galactosidase activity by *Bifidobacterium* decreases during weaning. Meanwhile, the resident *Firmicutes* increases which corresponds with changes in the relative abundance of major and minor species [117]. In another study, alteration in the gut microbiome of exclusively breastfed and non-breastfed infants before and after the introduction of solid foods revealed remarkable findings [39]. The identified differences revealed that breastmilk provides a stronger adaptability to the gut microbiome that eases the transition into solid foods. These findings clearly demonstrated the importance of maintaining supplementation of prebiotics during the weaning process.

Complementary feeding or the weaning stage combines the nutrition of newborns and young children during a time when their gut microbiome is in a very variable and unstable period. Starchy meals are frequently used as supplementary foods due to their texture and palatability. However, the health advantages of consuming starchy meals during this window of time remain uncertain. At this point in time, there have been no reports of any adverse effects or consequences associated with the use of starch during supplemental feeding. However, a key understanding that should be considered, should be the supplementation of prebiotics. Based on the growing evidence on the role of HMOs and other prebiotics on microbial structures, more recommendations must be given on the impact of the introduction of other types of prebiotics during the complementary feeding period. A suitable infant formula fortified with specific synthetic HMOs could be developed to support the transition to complementary feeding.

## 7. Conclusions

In conclusion, prebiotics are a very important component that contributes to the development of children from their early life. Newborns are fed with prebiotics, either naturally via breastmilk or artificially via formula. Prebiotic HMOs play a vital role in the development of the infant’s microflora, protection against infections, development of immune systems, and maturation of digestive systems. However, there are multiple factors including genetic and non-genetic factors that influence the composition of breastmilk, especially regarding HMOs. The total HMO composition of breastmilk was found to decrease over the lactation period due to changes in the regulatory mechanisms of the enzymes responsible for HMO synthesis. Additionally, there are also non-secretor mothers where their genetic factors have polymorphisms in HMO synthesizing genes, making them non-functional. This genetic polymorphism has no cure, and the non-HMO milk metabolome can have severe impacts on newborns’ health. Infants with non-secretor mothers were often found to have lower quantities of *Bifidobacterium* and increased pathogens. Although breastfed, the babies of non-secretor mothers experienced fewer health benefits as compared to those of secretor mothers. Therefore, infant formula fortified with synthetic HMOs such as 2′FL could be a promising innovation to improve infant’s nutrition of these non-secretory mothers. Further research is required to evaluate the bioactivity and bioavailability of these compounds when added to infant formulations in combination with other non-HMO prebiotics. Moreover, additional studies should be conducted to evaluate the physiological efficacy of the prebiotics that are still in the experimental phase when added to infant formula.

## Figures and Tables

**Figure 1 microorganisms-11-02453-f001:**
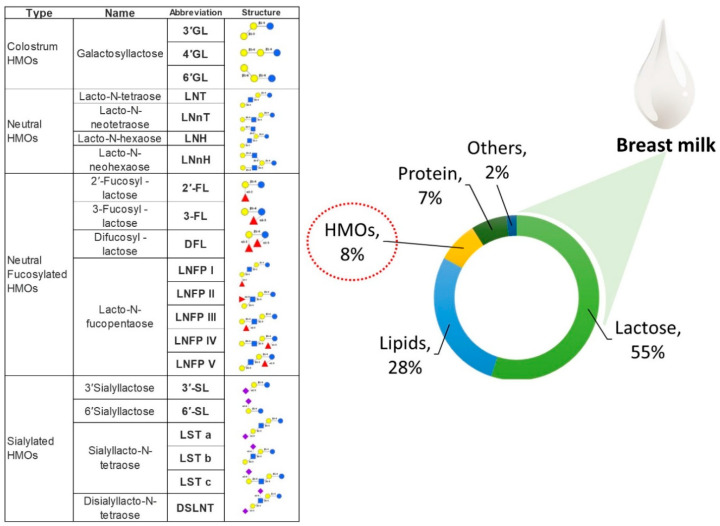
Composition of human breastmilk and types of human milk oligosaccharides.

## Data Availability

Not applicable.

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
