# Peer review of "Prebiotics in New-Born and Children’s Health"

_microorganisms, 2023, doi:10.3390/microorganisms11102453_

Round 1

Reviewer 1 Report

The authors have spent a lot of effort and collected a large amount of literatures, covering almost all researches related to human milk prebiotics. The review is rich in content and makes a good summary of previous such studies, which has good guiding significance for readers. However, the author merely lists the literatures, without corresponding summary and analysis from themselves. It is recommended to add summary and analysis at the end of each paragraph to give out the authors' own views. Are prebiotics in milk only HMOs? The main line of the paper jumps between prebiotics and HMOs. And the authors should further clarify and further refine the article to facilitate reading. The length of the paper is too long, and it needs to be shortened closely around the main line.

Minor editing of English language required

Reviewer 2 Report

Supplementation in newborns and infants, but also in adults, is still a very controversial topic. Therefore, well documented research is necessary and still missing. However functional food using microorganisms with prebiotic and probiotic properties is a trend of the future.This paper provides no evidence, but an article fits perfectly into the topic of the Special Issue Edition of  Microorganisms 2023children focusing on  Anti-inflammatory Property of Probiotics. The article is very extensive. It describes all the mechanisms of action of prebiotics in detail, requiring a lot of time on the part of the authors. The text is difficult to read due to this large amount of content. There are no underline points. However, it is understandable that for specialists and nutritional experts, this amount of basic information in one article is very important, even though there are many articles presenting all this important knowledge in a similar way. Individual fragments of the work concern: Prebiotics in the Early Life of Human Functionality of natural prebiotics in human milk Prebiotics in Infant Formula Milks. (Line 416 Typographical error Prebiotics in Infant Formula Milks” should be change to Prebiotics in Infant Formula Milk).Prebiotics during Weaning and Complementary Feeding

  Many of these topics have already been described. What is new in this information
about prebiotics. The authors should highlight it and whether there are any technological
innovations. The study does not provide   the clear criteria for how to use and to whom prebiotics.
should they be given to?
Recommending complementary feeding is certainly
a controversial novelty and it should be more detail described  and it must be included
in the conclusions.
The paper has a mixed form. It is partly a typical  descriptive
article presenting generally known knowledge regarding mainly breast milk
oligosaccharides and partly a systematic review of 17 publications. 
I propose to rewrite this work in the form only  as a  systematic review.
Table 1
shows the clinical interventions  using synthetic HMOs. This is very valuable part
of the work summarizing the most important clinically useful data.
The conclusions are
too long and are a summary of the information contained in the article and not the real
conclusion.
For the reviewer, the most important are the genetic mechanisms of action
of oligosaccharides due to individual differences in women.
We know that The amount
and diversity of HMOs are determined by the genetic background of the mothers
(HMO secretors or non-secretors). The non-secretor mothers secrete lower HMOs
than secretor mothers. The breastfed infants of secretor mothers gain more health
benefit than those of non-secretor mothers. So maybe for non secretors mothers 
supplementation of infant formula with 2′-FL and LNnT is a promising innovation
for infant nutrition.
However the authors do not comment on this issue.
This issue in my opinion can be the most important conclusion.  The work should
contain some the authors' own experiences, som views and thoughts on this topic.
               Below I present my comments and questions for the authors and a request
for clarification.
Line 50 “Breast milk is the recommended nutrition for infants, but for various reasons,
breast feeding is not always possible”
 This is true, but there are very few
contraindications to breastfeeding and it should be emphasized once again that
  the
huma
milk oligosaccharides (HMOs) are the third most abundant component of breast
milk, after lactose and lipids
. HMOs in the human breast milk are a complex mixture
of more than 200 non-digestible and non-nutritional carbohydrates.
Line 707  “In 6 to 8 months of life, the infant’s digestive systems are uniquely suited
to digesting the macronutrient elements found in breast milk and infant formula  milk
[41]. Thus, there is a growing recommendation for the use of staged and follow-on
formulae, which exhibit variations in composition based on the necessary daily
allowances and the introduction of complementary meals “.
At this point, there should
be a clear encouragement by the authors to breastfeed as long as possible. Why add
formula when  we  mother starts complementary feeding ????? This requires explanation.
Line 715 “Therefore, during the weaning process, supplementation of prebiotics in solid foodsmust be continued to establish and maintain a healthy gut
microbiome in  children.
  I think that if a mother breastfeeds for a long time, the gut microbiome is already properly formed. Early weaning may be an indication in some for the use of prebiotics. however, this should be properly highlighted. Line 716-718  “Additionally, several health organizations including the WHO
recommend  breastmilk must be accompanied by complementary feeding. This clarifies
the importance  of continuing supplementation of prebiotics in infants besides, aiding
the digestion
”. This explanation is not understandable. After all, mothers often feed
their children until they are 2 years old.
How long prebiotic supplementation is
suggested by the authors
and why. The child should receive proper nutrition and has
breastfed for as long as both  (mother and child) of them want it.
Are the figures the original contribution of the authors? If not, why is the source not
provided and the editor of the 
Microorganisms has written consent from the original
authors to use it?
In general, the work contains too few aspects of the benefits of breastfeeding.
